# Marked Presence of Methicillin-Resistant *Staphylococcus aureus* in Wild Lagomorphs in Valencia, Spain

**DOI:** 10.3390/ani10071109

**Published:** 2020-06-29

**Authors:** Elena Moreno-Grúa, Sara Pérez-Fuentes, David Viana, Jesús Cardells, Víctor Lizana, Jordi Aguiló, Laura Selva, Juan M. Corpa

**Affiliations:** 1Biomedical Research Institute (PASAPTA-Pathology group), Facultad de Veterinaria, Universidad Cardenal Herrera-CEU, CEU Universities, C/Tirant lo Blanc 7, 46115 Alfara del Patriarca, Valencia, Spain; elena.moreno3@uchceu.es (E.M.-G.); saraperezfue@gmail.com (S.P.-F.); dviana@uchceu.es (D.V.); 2Servicio de Análisis, Investigación, Gestión de Animales Silvestres (SAIGAS), Facultad de Veterinaria, Universidad Cardenal Herrera-CEU, CEU Universities, C/Tirant lo Blanc 7, 46115 Alfara del Patriarca, Valencia, Spain; jcardells@uchceu.es (J.C.); victor.lizana@uchceu.es (V.L.); jordi.aguilo@uchceu.es (J.A.); 3Wildlife Ecology & Health group (WE&H), Universitat Autònoma de Barcelona (UAB), Edifici V, Travessera del Turons, 08193 Bellaterra, Barcelona, Spain

**Keywords:** methicillin-resistant *Staphylococcus aureus*, MRSA, *mec*C gene, wild rabbits, high-density areas

## Abstract

**Simple Summary:**

The presence of bacteria resistant to antibiotics, such as *Staphylococcus aureus* resistant to methicillin (MRSA), is becoming an increasing everyday concern for their implications for human and animal health. This is even more alarming when such bacteria are isolated in wild animals which, in principle, should not come into contact with antibiotics. This work studied 353 rabbits and 10 hares hunted in rabbit high-density areas. Of these, 41.3% carried *S. aureus* in some sampled locations, of which 63.3% were MRSA. These are surprising results given the unexpected high presence of MRSA in the studied animals. This finding is very worrying because these animals tend to enter the food chain with no veterinary control, which implies a risk for human health. Thus it is necessary to extend this study to other, less-populated areas with other animal species (ruminants, rabbit predators, hunting dogs, etc.) or even water sources to obtain further knowledge about the origin of bacterial resistances in nature.

**Abstract:**

The appearance of methicillin-resistant strains of *Staphylococcus aureus* (MRSA) in several animal species (including rabbits) has set off alarms for their capacity to act as reservoirs for this bacterium. This is especially important in wild animals given its epidemiological implications. The objectives of this study were to identify and characterize *S. aureus*, specifically MRSA, strains in wild lagomorph high-density areas. Ten hares and 353 wild rabbits from 14 towns with a high rabbit density in the Valencian region (eastern Spanish coast) were sampled. Swabs from the nasal cavity, ears, perineum and lesions (when present) were taken for microbiological studies. The detection of different genes and antibiotic susceptibility studies were also carried out. Of all the animals, 41.3% were positive for *S. aureus*, of which 63.3% were MRSA. Ears were the anatomical location with more *S. aureus* and MRSA strains. The more frequently identified MLST type was ST1945 (97.1%, 136/140). The *mec*A gene was found only in one sample. The rest (*n* = 139) carried the *mec*C gene and were included in CC130, except one. Penicillin resistance was detected in 28 *mec*-negative isolates and, in one case, bacitracin resistance. *mec*A isolate presented resistance to enrofloxacin and tetracycline, and 10 *mec*C isolates also showed bacitracin resistance. No MRSA isolate was positive for genes *chp*, *sea*, *tst* and PVL. Two ST1945 isolates contained IEC type E (comprising genes *scn* and *sak*). *mec*A-isolate was positive for *bla*Z. Of the 28 MSSA strains showing resistance to penicillin, 22 carried the *bla*Z gene. These surprising results highlight the marked presence of MRSA strains in wild rabbits in high-density areas.

## 1. Introduction

*Staphylococcus aureus* is a widely distributed bacterium in nature and is often considered a frequent host of skin [1] and mucous, mainly in the nasal cavity [2]. Studies have shown that about 20% of humans are persistent nasal carriers of *S. aureus*, and around 30% are intermittent carriers [2]. The percentage of nasal carriers reported in different animal species varies: 7.9% in horses [3], 29% in ewes [4] and 32.1–53% in rabbits [5,6]. Asymptomatic nasal carriers play a key role in the epidemiology and control of staphylococcal diseases, as the nasal cavity allows bacteria to persist over time and to multiply, which constitutes a source of infection [2]. In human medicine, the main problem caused by *S. aureus* occurs in hospitals, with the most important cause of not only nosocomial infections [7], but also community-acquired infection [8]. *S. aureus* is also a major pathogen in veterinary medicine that affects various animal species. In commercial rabbits, it has been signaled as one of the main causes of culling on farms [9,10]. In these cases, staphylococcal infections in rabbitries are caused by the international dissemination of the ST121 lineage of *S. aureus*, and other less frequent lineages like ST96 [11,12].

The increase in bacterial resistance to antibiotics in recent years has become a serious health problem. *S. aureus* is perfectly capable of acquiring multiple resistance mechanisms to several antimicrobial agents [13,14], which limits their therapeutic effectiveness. One of the most important ones for its clinical repercussions is resistance to methicillin. Methicillin-resistant *Staphylococcus aureus* (MRSA) evolved from Methicillin-susceptible *Staphylococcus aureus* (MSSA) by acquiring SSC*mec* elements containing a *mec* gene (*mec*A, *mec*C), which encodes a protein with a low affinity for β-lactam antibiotics [15]. 

In animals, livestock-associated MRSA (LA-MRSA) emerged in pigs in 2005 [16] and was later described in other animal species. The most widely related clonal complex (CC) to LA-MRSA is CC398. This CC was also isolated from farmers who had been in close contact with infected animals [17], which indicates that this clone plays an important role as a reservoir of transmission to humans [18]. Therefore, special attention has been paid to the colonization of animals with *S. aureus* because they may potentially act as a reservoir to humans [15,19,20]. 

Rabbits are among the animal species in which LA-MRSA CC398 strains have been isolated. The first case in rabbits was reported on a commercial farm in Italy, which also involved farm workers and their families [21]. Recently, LA-MRSA strains have been described to belong to ST2855 (CC96), ST146 (CC5), ST398 (CC398) and ST4774 (CC130) from several rabbitries of the Iberian Peninsula [22]. These CC have been related to illness in humans [18,23,24,25,26].

MRSA strains have been isolated in human hospitals and on animal farms where antibiotics are regularly used. These antibiotic-resistant organisms can spread to communities and the environment [27]. Therefore, free-living animals might be colonized or infected by human and livestock sources, and can be associated with contaminated environments, even though they do not directly come into contact with antimicrobial drugs [28].

In recent studies, a high *S. aureus* carriage rate has been detected in wild animals, including European wild rabbits. The most relevant finding was that all the isolates were MRSA [29], which suggests a wildlife MRSA reservoir. Additionally MRSA, belonging to CC130, has been detected in diseased European brown hares (*Lepus europaeus*) [30,31].

The wild rabbit (*Oryctolagus cuniculus*) is an extremely abundant endemic species of the Iberian Peninsula in some areas, where conservation of endangered predators is essential, but where major crop damage is a problem [32]. Rabbits and wild hares are also subject to hunting and are subsequently consumed domestically, often with no adequate veterinary control or sanitary management.

The role of *S. aureus* as a possible pathogen or colonizer in wild populations of lagomorphs has not yet been studied in a massive and systematic manner. In high-density areas, the interaction between rabbits and other animals is higher and, therefore, the probability of *S. aureus* transmission will also be higher. For these reasons, this study sets out the following objectives: (1) to know the role of wild rabbits and hares as a reservoir of *S. aureus* in wild lagomorph high-density areas; (2) to characterize the *S. aureus* strains isolated from wild rabbits and hares to compare them with those obtained previously on commercial farms; (3) to study the prevalence of MRSA strains in wild populations.

## 2. Materials and Methods 

### 2.1. Sampling, Isolation and Characterization of S. aureus Isolates

#### 2.1.1. Animals and Geographical Locations

Ten hares (*Lepus granatensis*) and 353 wild rabbits (*Oryctolagus cuniculus*) from one game range or more located in 14 high-density towns were sampled to check the presence of *S. aureus*. The 10 hares of the study were included because they were hunted together with rabbits and it was considered interesting to compare the findings observed in another lagomorph from the same geographical locations. Animals were donated by hunters for this study once they were dead. The study was carried out in the Valencian Region, formed by the provinces of Valencia, Castellon and Alicante (each, in turn, formed by several districts) in eastern Spain (Figure 1) in 2019.

The lagomorph high-density towns were defined by the Valencian Regional Government (Order of June 11, 2009 and updated by Resolution of November 20, 2018 [32]). The Valencian Regional Government, in collaboration with professional agricultural organizations, conducts surveys to identify those areas in different regions where the inordinate proliferation of rabbit populations associated with crop damage has been observed, and where hunting is allowed all year long. 

The regional Valencian districts were classified as: extreme density (>70 rabbits/100 Has/year), very high density (31–70 rabbits/100 Has/year), high density (21–30 rabbits/100 Has/year), medium density (10–20 rabbits/ 100 Has/year) and low density (<10 rabbits/100 Has/year). This classification was based on the last 5-year (2012–2017) declared annual captures (rabbits per 100 hectares) published by the Fish and Game Service of the Valencian Regional Government [33] (Figure 1).

Samples from the towns of Pedralba and Vinarós were included in this study, which belong to low- and medium-density districts, respectively, and are located on the borders of very high- or high-density districts, respectively, that also host high densities due to the proximity to these districts.

In order to calculate the necessary representative amount of rabbits to be sampled, the average district hunting bag of the last 5 years (2012–2017) was taken as the population data. The premise of an unknown prevalence in the *S. aureus* carrier condition was included in the analysis (WinEpi 2.0, Zaragoza, Spain). Such a restrictive criterion meant that the initially established number was not reached in several districts, namely El Baix Maestrat, El Camp de Túria, La Plana Utiel-Requena, La Costera and La Vall d’Albaida, as they lacked catches during the study period.

#### 2.1.2. Microbiological Studies

The studied animals were referred to the Universidad CEU Cardenal Herrera (Alfara del Patriarca, Valencia, Spain), where necropsy and sampling were carried out. Swabs from the nasal cavity, ears, perineum and lesions (whenever present) were taken. 

Samples were inoculated on blood-agar (Becton–Dickinson, Sparks, MD, USA) and incubated aerobically at 37 °C for 24–48 h. *S. aureus* strains were identified on the basis of morphological growth characteristics and hemolytic properties [34]. When contamination appeared with other bacteria, Mannitol Salt Agar (Becton–Dickinson, Sparks, MD, USA) was used to obtain isolated *S. aureus* colonies. Only the plaques with 10 or more *S. aureus* CFU were considered positive. Those colonies compatible with *S. aureus* were grown in TSB (Tryptic Soy Broth) at 37 °C for 12 h with shaking. To perform PCRs, genomic DNA was extracted from each isolate by a Genelute Bacterial Genomic DNA kit (Sigma-Aldrich, St. Louis, MO, USA) according to the manufacturer’s protocol, except for bacterial cells, which were lysed by lysostaphin (12.5 μg/mL, Sigma-Aldrich) at 37 °C for 1 h before DNA purification. Isolates were genotyped based on the analysis of the polymorphic regions of genes *coa* and *spa* as previously described [35]. Multilocus-sequence typing (MLST) was performed in the selected isolates [36]. According to sequence type (ST), isolates were ascribed to the different clonal complexes (CC). CC were assumed for some isolates according to their specific genotype [22]. 

All the strains were checked for the presence of *mec*A/*mec*C genes by PCR (Table 1), as previously described [37,38]. Isolates were classified into MRSA or MSSA according to the presence or absence results of the *mec*A/*mec*C genes obtained by PCR.

#### 2.1.3. Detection of *chp*, *sak*, *scn*, *bla*Z, *tst*, *sea*, and PVL-encoding Genes, and *agr* Typing

An MRSA strain of each ST type was selected from all the towns. These selected MRSA isolates were subjected to a PCR assay (Table 1) to detect the *lukF*/*S-PV* genes encoding the PVL toxin, and genes *sea* and *tst* that encode the SEA and TSST-1 toxin and *agr* typing, as previously described [39]. Strains were also checked for the presence of immune-evasion cluster (IEC) genes (*sak*, *chp*, and *scn*) by PCR, as described elsewhere [40]. *bla*Z gene detection was investigated by PCR [41].

#### 2.1.4. Antibiotic Susceptibility Testing

The antibiotic susceptibility of the all isolates was determined by the disk diffusion method on Mueller–Hinton agar (Becton–Dickinson, Sparks, MD, USA) according to the recommendations of the Clinical and Laboratory Standards Institute (CLSI). The disk diffusion assay was run with 15 antibiotics: bacitracin (10 U), enrofloxacin (5 μg) (OXOID), streptomycin (10 μg), spiramycin (100 μg), sulfadiazine (25 μg), chloramphenicol (30 μg) (BD), doxycycline (30 μg), erythromycin (15 μg), gentamicin (10 μg), neomycin (30 μg), penicillin (10 U), tetracycline (30 μg), cefoxitin (30 μg), trimethoprim/sulfamethoxazole (1.25 μg/23.75 μg, respectively), and vancomycin (30 µg and 5 µg) (BIO-RAD, Hercules, CA, USA). *S. aureus* strain ATCC 25923 and *Enterococcus faecalis* strain ATCC 29212 were used as controls in the susceptibility test.

### 2.2. Statistical Analysis

In order to know if the number of animals positive for *S. aureus* was statistically representative in relation to the number of studied animals in each district, the following statistical analysis was carried out. Categorical data were compared by the Fisher exact test. All the reported *p* values were two-tailed and analyses were performed using the GraphPad software. The variables with *p* < 0.01 were considered statistically significant.

## 3. Results

### 3.1. Identification of S. aureus, MRSA and Lesions

*S. aureus* was isolated in 41.3% (150/363) of all the wild rabbits and hares included in this study in at least one of the sampled locations (nasal cavity, ears or perineum). Of the 150 animals positive for *S. aureus*, the *mec*C gene was detected in 94 (62.7% of the positive animals) and the *mec*A gene was observed in one animal (0.7% of the positive animals) (Table 2). 

Of the 10 hares included in this study, only one from the town of Cheste carried *S. aureus* in its nostrils. The strain isolated from this hare was MSSA, and it had a different CC (CC121) to the isolated strains of rabbits hunted in the same district (CC5, CC130, CC398 and CC425).

The number of animals carrying *S. aureus* varied among the different sampled towns (Table 2). The towns with more animals carrying *S. aureus* were Pedralba (*n* = 37), Castellón de Rugat (*n* = 26) and Alfafara (*n* = 19). In addition, Alfafara, Pedralba and Cheste were the towns where a high percentage of animals with MRSA was detected (100%, 86.5% and 69.2%, respectively) and they all carried the *mec*C gene, except for one from the town of Cheste for which *mec*A was identified.

In all the following were found: 244 *S. aureus* isolates (244/1100; 22.2%) from the swabs taken from the nasal cavity, ears and perineum (*n* = 1089) in the 363 animals; additional swabs of six rabbits and two hares presenting one lesion or more (*n* = 11); 57.4% (140/244) of the *S. aureus* isolates were MRSA (Table 3). Of the 140 MRSA isolates, *mec*A was detected in only one isolate, and *mec*C was amplified in the 139 remaining isolates. 

When considering the anatomical location of sampling, ears were by far the place from which more positive samples were isolated and where a higher percentage of MRSA was found. *S. aureus* was isolated in 33.3% (121/363) of the swabs taken from ears, and 66.1% of them were MRSA (80/121). In nasal cavity and perineum, fewer swabs were positive for *S. aureus* (70 and 45, respectively), but MRSA strains were also detected (48.6% and 44.4%, respectively) (Table 3). In general, very few lesions were observed after the necropsy of animals, except for traumatic lesions caused by animal hunting with firearms. However, *S. aureus* had evolved in many of them because of the 11 samples taken from lesions (dermatitis, hepatic abscesses, conjunctivitis), eight were positive for *S. aureus* and six carried the *mec*C gene.

In some animals, *S. aureus* strains were isolated in different locations at the same time. Therefore, the number of animals carrying *S. aureus* and the number of positive samples differed. Eighty-five animals carried *S. aureus* only in one location; in 45, 16 and 4 rabbits, the bacterium was isolated in 2, 3 or 4 different locations, respectively. Another interesting result was that two different *S. aureus* strains were found in the same anatomical location in five animals.

### 3.2. Characterization of S. aureus Isolates

The MLST typing analysis revealed 13 different STs (Table 4). The MLST type with the most isolates was ST1945 (*n* = 177), followed immediately by ST425 *(n* = 42) and ST121 (*n* = 7). Other less frequent STs were ST5821 (*n* = 4), ST5826 (*n* = 3), ST398 (*n* = 3), ST5 (*n* = 2) and ST5822, ST5823, ST5824, ST5825, ST5844 and ST5845 (*n* = 1, each).

Most of the strains isolated from animals belonged to CC130 (74.2%; 181/244). The next most prevalent CC was CC425 (19.3%; 47/244). The other found CCs were CC121 (7/244), CC398 (3/244) and CC5 (2/244), but they were infrequently isolated. 

The 140 MRSA isolates belonged mostly to ST1945 (97.1%, 136/140). The remaining four isolates corresponded to ST398, ST5822, ST5823 and ST5824. The *mec*A gene was found only in one sample, the only MRSA belonging to ST398 (CC398). The other samples (*n* = 139) carried the *mec*C gene and were included in CC130, except for one (ST5824), which was a CC that has not yet been described.

Four rabbits presented lesions and carried *S. aureus* in some anatomical locations simultaneously. The genotypic analysis showed that the *S. aureus* strains isolated from nostrils/ear/perineum and lesions were clonally related in 100% animals. In a single case, corresponding to OC19006, a strain different from those from rostril/perineum was identified in the lesion (Table 5).

### 3.3. Antibiotic Resistance Profile

Antibiograms were performed of all the *S. aureus* positive isolates obtained in each studied town (244 isolates: 140 MRSA and 104 MSSA). Penicillin resistance was detected in 28 *mec* negative isolates and, in one case, bacitracin resistance. *mec*A isolate presented resistance to enrofloxacin and tetracycline, and 10 *mec*C isolates also showed bacitracin resistance. Of the 140 MRSA strains, 49 were sensitive to cefoxitin and 34 to penicillin. None of the 104 MSSA strains showed resistance to cefoxitin, but 28 strains resisted penicillin. Eleven isolates were Bacitracin-resistant (10 MRSA and 1 MSSA). The strain isolated with the *mec*A gene was also resistant to enrofloxacin and tetracycline (Table 6). 

### 3.4. Detection of the IEC Cluster (scn, chp, sak and sea), blaZ, tst and the PVL Genes among MRSA Isolates

The PCR detection of the IEC genes, *bla*Z, *tst*, and the PVL-encoding genes was carried out to select MRSA isolates. None of these isolates were positive for genes *chp*, *sea*, *tst* and PVL. Two ST1945 isolates contained the IEC type E (comprising genes *scn* and *sak*). The *mec*A-isolate was positive for *bla*Z (Table 6). Of the 28 MSSA strains displaying resistance to penicillin, 20 carried the *bla*Z gene. Figure 2 shows the positive and negative controls of the genes tested in this study, and the size of the PCR amplified fragments of each gene.

## 4. Discussion

In the present study, 244 *S. aureus* isolates obtained from 150 wild rabbits and hares, located in 10 high-density population areas (districts) of the Valencia Region in Spain, were analyzed. As far as the authors are aware, this is the first time that a study with such a large number of animals, conducted in high-density areas of animals, has been carried out. The reason why this study was performed in rabbit high-density areas was because the probability of detecting *S. aureus* carrier animals was believed to be higher in these areas. In total, 41.3% (150 of 363) of the animals carried *S. aureus* and positive animals were detected in all the studied geographical areas. Apparently, there was no clear geographical distribution of towns with more positive animals, although they were mainly located in the center and south (La Vall d’Albaida and El Comtat) of the Valencian region. Further studies in areas with lower rabbit density would be necessary to assess the relationship between *S. aureus* prevalence and rabbit density of animal.

Sixty-five animals carried *S. aureus* in more than one location; and two different *S. aureus* strains were identified in the same anatomical location in five rabbits. This agrees with previous studies which have reported how several different *S. aureus* strains can simultaneously colonize individual rabbits [42]. This is an important issue because it could affect the diagnosis of this condition if only one colony is selected during the microbiological identification of staphylococcal infections or only one anatomical site is selected for sampling

In humans, the nose is the main ecological niche where *S. aureus* resides [2]. However, in this study, the highest percentage of *S. aureus* was detected in ears (49.6%). This finding has also been observed in commercial rabbitries where the presence of *S. aureus* was isolated from mainly samples taken from the ears and perineum among nine different anatomical locations [42]. The percentage of nasal carriers (28.7%) was lower than previously described in farm rabbits with staphylococcal problems (56%) [6], but was higher than other wild mammals (22.3%), and was substantially higher than recently described for wild rabbits (8%) [29]. 

In the present study, only eight animals (6 rabbits, 2 hares) showed lesions; *S. aureus* was isolated from the lesions presented by the 6 rabbits. The high percentage of samples (8 of 11) obtained from the animals positive for *S. aureus* (72.7%) was surprising, especially when considering the different characteristics of the observed lesions compared to commercial rabbits [43]. Four of the six animals with positive *S. aureus* lesions (66.7%) also carried this bacterium in their noses. In humans, it is reported that nasal carriers of *S. aureus* are at increased risk of acquiring infection with this pathogen [2]. This has also been found in animals, specifically rabbits, where the colonization capacity of this bacterium plays an important role in spreading the disease [42]. There are reports that *S. aureus* carriage in rabbits can be a risk for developing clinical infections [6]. These results agree with this asseveration because the *S. aureus* strains isolated from nostrils/ear/perineum and lesions were clonally related in 100% animals. Only in one case, corresponding to rabbit OC19006, was a strain different from those from rostril/perineum identified in the lesion.

In order to understand the diffusion and possible origin of *S. aureus* (especially MRSA), correctly identifying the involved strain is vital. The most frequently detected CC herein was CC130, followed by CC425. The ST121 (CC121) lineage has been identified in most nasal carriers [6] and chronic staphylococcal infections [12,22,44] in commercial rabbits. Therefore, these results indicate that the *S. aureus* strains which affect commercial and wild rabbits are different, which reinforces recently observed findings in wild rabbits in Aragón (a region in north Spain), where three strains were typed as t843 (ascribed to CC130). This finding has also been observed in samples from other wild species, such as reed deer (*Dama dama*) and wild boar (*Sus scrofa*) [29]. Only one hare carried *S. aureus*, isolated from its nostril (MSSA), and it differed (CC121) from the strains isolated in the rabbits from the same district (CC5, CC130, CC398 and CC425). However, this number was not big enough to collect data on possible species specificity.

In the present study, 138 of the 181 CC130 strains carried *mec*C and the rest were methicillin-sensitive. The isolates reported to date carrying *mec*C belong mainly to common lineages in cattle, namely CC130, CC1943, and CC425, which suggests a zoonotic reservoir [19]. Besides cattle, *mec*C has been found in other farm animals, with isolates ST130 in sheep [45,46], ST4774 (a novel tpi single locus variant of ST130) in rabbits [22] and isolate ST425, which caused highly virulent infection, in one rabbit [47]. ST425 is a lineage that has been found in both wild and domestic animals, and has been previously noted in wild boar from Germany [48], red deer from Spain [49] and also in humans. In this work, all the ST425 strains were methicillin-sensitive. The appearance of ST425-MRSA-XI, from cattle in the UK in 2011 [50], in wildlife (fallow deer, wild boar), and in an environmental sample from Spain [51], should put us on alert and to follow up these lineages in the future. 

One interesting finding was that the only strain to carry the *mec*A gene belonged to the ST398 lineage. The first LA-MRSA case reported in rabbits for meat production belonged to this lineage [21]. Moreover, the first case of LA-MRSA in rabbits in Spain showed limited genetic diversity (ST2855, ST146, ST398, ST4774), with ST2855 being the predominant clone [22]. It is notorious that this study found neither this lineage nor ST146 strains. This ST belongs to CC5, which has been previously described in *S. aureus* isolates from rabbit carcasses, but were not resistant to methicillin [52].

Of all the strains, 72.5% belonged to the ST1945 lineage, which is included in CC130. ST1945, a single-locus variant of ST130, has been reported as an *mec*C-carrying MRSA in humans in the UK [50], Germany [53], France [54] and Spain [55]. This lineage has been described in fecal samples taken from free-ranging wild small mammals in southern Spain, specifically in two wood mice. MRSA were detected in 7% of the analyzed wood mice [56]. The MRSA percentage herein observed was much higher, as 57.4% of *S. aureus* isolates had MRSA. 

Another technique used to characterize the MRSA strains included the identification of the *agr* and staphylococcal cassette chromosome *mec* (SCC*mec*). In our study, all the isolates belonged to *agr* type III, as reported in other series [53], except for isolate ST398, which was *agr*I. CC130 presented SCC*mec*XI and ST398 SCC*mec*V, as previously described [50,57,58].

Two *mec*C-positive MRSA isolates ST1945 carried genes *scn* and *sak* from the IEC system, and were consequently ascribed to IEC-type E. The origin of the *mec*C gene is unclear, but it has been detected in staphylococci from humans and animals [56]. Very little research has determined the presence of IEC genes in *mec*C-positive isolates, and all the available research works found isolates lacking genes *sak*, *chp* and *scn* [31,53,59], which supports the hypothesis of the possible animal origin of these isolates. The detection of genes *sak* and *scn* in these two strains is relevant and poses questions about the potential origin of these isolates.

The possible origin of the MRSA strains remains an interesting enigma. In Spain, some wild animals (wild boars and fallow deer) located on a game estate had similar *mec*C-MRSA isolates detected in river water. Therefore, it was proposed that water could be a source that disseminates this type of strain in nature [51]. The colonization of wild animals with *S. aureus* lineages from humans has been confirmed in a zoological park [60]. The transmission of MRSA strains between animals and persons is relatively easy, because it has been observed how 22% of students who visited pork farms (30% prevalence of MRSA) became MRSA nasal carriers [18]. It has also demonstrated that the most prevalent *S. aureus* strain in commercial rabbits in Spain (ST121) has a human origin. This host adaptation took place 45 years ago when a single nucleotide mutation was sufficient to convert a human-specific *S. aureus* strain into one that could infect rabbits [61]. However, close contact between humans and animals has been assumed in all these cases, and the animals on commercial farms lived in high-density communities with interactions between animals. These conditions are not, a priori, those that take place in wild lagomorphs. However, it is worth considering that the present study entails the peculiarity of being conducted in rabbit high-density areas that could increase the probability of contact between animals and favor *S. aureus* spreading through rabbit populations. Additionally, CC130 (the most isolated CC in this study) is a lineage associated mainly with cattle, with transmission patterns that might be assumed in which wild rabbits could be infected by ingesting contaminated feces from wild ruminants in the same area. Our study did not analyze water or other animals that could interact with wild rabbits (predators, ruminants, etc.). Therefore, it would be interesting to extend this study along this line in the future to know the epidemiological origin of these bacteria.

Most MRSA isolates show resistance to cefoxitin. As previously reported for *mec*C-carrying MRSA, most isolates were susceptible to all the tested antibiotics, except for β-lactams [55], and 10 strains were Bacitracin-resistant. However, the ST398-MRSA-*mec*A strain also shows resistance to enrofloxacin and tetracycline, and carried the *bla*Z gene. Resistance to tetracycline has also been associated with livestock-related ST398 [62]. 

Food-producing animals, both livestock and wildlife, and derived products are considered potential sources of MRSA in humans [63]. However in that study, which included 1365 wild animals, *S. aureus* was detected in only 2.0% of wild animal carcasses and in 3.2% of wild boar lymph nodes, and none had MRSA. Therefore, the authors concluded that the risk of transmission to humans was limited. Conversely, in our study, the high presence of *S. aureus* (41.3% of animals), of which 63.3% had MRSA, indicates an alarming situation for wild rabbits in our study area, because these animals can be consumed directly by hunters, or even be donated to charities, without undergoing adequate veterinary inspections.

## 5. Conclusion

In conclusion, the results suggest that wild lagomorphs can constitute a reservoir of MRSA isolates in nature. Our report confirms the high presence of the MRSA CC130 lineage containing the *mec*C gene in wild rabbits and hares in east Spain, which could be transmitted to domestic animals or humans with major public health implications. It would be advisable to conduct similar studies in low-density areas of animals and in other regions of Spain and Europe to determine the degree of dissemination of MRSA isolates in lagomorphs and other wild species.

## Figures and Tables

**Figure 1 animals-10-01109-f001:**
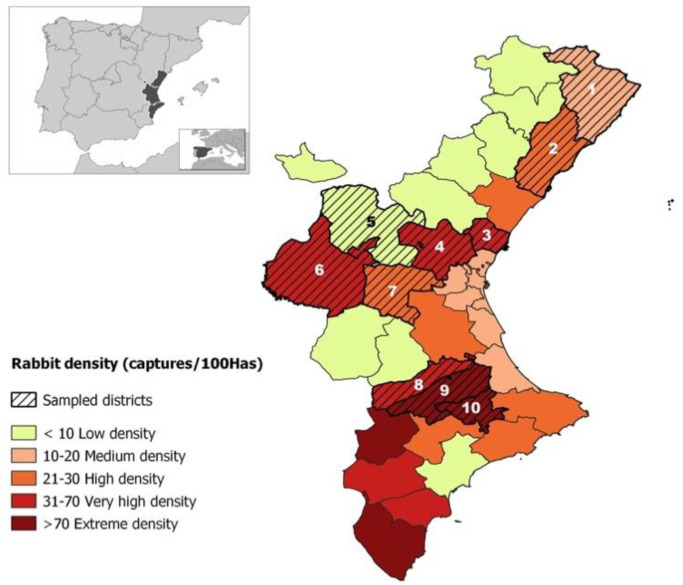
Districts of the Valencian Region (El Baix Maestrat (1) La Plana Alta (2), Camp de Morvedre (3), El Camp de Túria (4), Serranos (5), La Plana Utiel-Requena (6), Hoya de Buñol (7), La Costera (8), La Vall d’Albaida (9), El Comtat (10)), colored according to rabbit density.

**Figure 2 animals-10-01109-f002:**
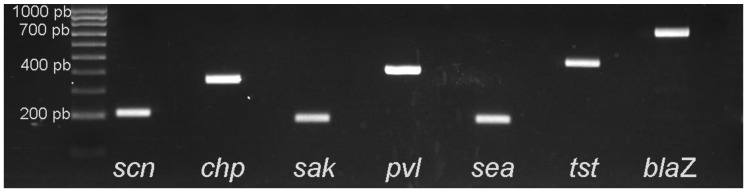
PCR detection of the IEC Cluster (scn, chp, sak and sea), blaZ, tst and the PVL genes among MRSA isolates. The figure shows the positive and negative control of each gene.

**Table 1 animals-10-01109-t001:** The primer sequences used in this study.

Primer		Sequence (5′–3′)	References
*mec*A	*mec*A-1m	GTAGAAATGACTGAACGTCCGATAA	[37,38]
	*mec*A-2c	CCAATTCCACATTGTTTCGGTCTAA	
*mec*C	*mec*C-1m	CTACACGTTCCATACCATTAG	[37,38]
	*mec*C-2c	CGCCTCTATGATAAACAATTGC	
*lukF*/*S-PV*	*lukF*/*S-PV*-1m	ATCATTAGGTAAAATGTCTGGACATGATCCA	[39]
	*lukF*/*S-PV*-2c	GCATCAAGTGTATTGGATAGCAAAAGC	
*sea*	*sea*-1m	AAAGTCCCGATCAATTTATGGCTA	[39]
	*sea*-2c	GTAATTAACCGAAGGTTCTGTAGA	
*tst*	*tst*-1m	CTAATCAAATAATCAAAACTGC	[39]
	*tst*-2c	TTTCCAATAACCACCCGTTT	
*agr* type I	*agr* type I-1m	GTCACAAGTACTATAAGCTGCGAT	[39]
	*agr* type I-2c	ATGCACATGGTGCACATGC	
*agr* type II	*agr* type II-1m	TATTACTAATTGAAAAGTGGCCATAGC	[39]
	*agr* type II-2c	ATGCACATGGTGCACATGC	
*agr* type III	*agr* type III-1m	GTAATGTAATAGCTTGTATAATAATACCCAG	[39]
	*agr* type III-2c	ATGCACATGGTGCACATGC	
*agr* type IV	*agr* type IV-1m	CGATAATGCCGTAATACCCG	[39]
	*agr* type IV-2c	ATGCACATGGTGCACATGC	
*sak*	*sak*-1m	AAGGCGATGACGCGAGTTAT	[40]
	*sak*-2c	GCGCTTGGATCTAATTCAAC	
*chp*	*chp*-1m	TTTACTTTTGAACCGTTTCCTAC	[40]
	*chp*-2c	CGTCCTGAATTCTTAGTATGCATATTCATTAG	
*scn*	*scn*-1m	ACTTTAGCAATCGTTTTAGC	[40]
	*scn*-2c	CTGAAATTTTTATAGTTCGC	
*bla*Z	*bla*Z-1m	CAGTTCACATGCCAAAGAG	[41]
	*bla*Z-2c	TACACTCTTGGCGGTTTC	

**Table 2 animals-10-01109-t002:** Animals positive for *S. aureus* and MRSA in the different studied districts.

Id	District	Town	No. of Animals Tested	No. of Animals *S. aureus* Positive ^1^	*p* ^2^	Animals *mec*C+	Animals *mec*A+	*mec* (%)
1	El Baix Maestrat	Vinaroz	1	1	0.4132	1	0	100
2	La Plana Alta	Cabanes	120	18	0.0001	10	0	55.6
3	Camp de Morvedre	Faura	31	9	0.1826	1	0	11.1
4	El Camp de Túria	Llíria	1	0	1	0	0	0
4	El Camp de Túria	Vilamarxant	2	1	1	1	0	100
5	Serranos	Pedralba	39	37	0.0001	32	0	86.5
6	La Plana Utiel-Requena	Requena	2	0	0.5137	0	0	0
6	La Plana Utiel-Requena	Utiel	32	6	0.0077	0	0	0
7	Hoya de Buñol	Cheste	53	13	0.0098	8	1	69.2
7	Hoya de Buñol	Godelleta	15	11	0.0143	4	0	36.4
8	La Costera	Montesa	1	0	1	0	0	0
9	La Vall d’Albaida	Castelló de Rugat	28	26	0.0001	15	0	57.7
9	La Vall d’Albaida	Montaverner	18	9	0.4695	3	0	33.3
10	El Comtat	Alfafara	20	19	0.0001	19	0	100
Total			363	150		94	1	63.3

^1^ Number of animals positive for *S. aureus* at one or more of the sampling sites. ^2^ Values in bold denote the variables with *p* < 0.01, considered statistically significant.

**Table 3 animals-10-01109-t003:** Positive samples to *S. aureus* and MRSA according to sampling localization (in the different studied locations).

Sampling Localization	Samples	*S. aureus*	(%)	MRSA	(%)
Nasal cavity	363	70	19.3	34	48.6
Ears	363	121	33.3	80	66.1
Perineum	363	45	12.4	20	44.4
Lesions	11	8	72.7	6	75
Dermatitis	6	5	83.3	3	60
Hepatic abscesses	2	2	100	2	100
Conjunctivitis	3	1	33.3	1	100
Total	1100	244	22.2	140	57.4

**Table 4 animals-10-01109-t004:** Correlation between MLST and the clonal complexes of the studied isolates.

MLST	CC	MRSA	MSSA	Total
ST1945	CC130	136	41	177
ST425	CC425	-	42	42
ST121	CC121	-	7	7
ST5821	CC425	-	4	4
ST5826	singleton	-	3	3
ST398	CC398	1	2	3
ST5	CC5	-	2	2
ST5822	CC130	1	-	1
ST5823	CC130	1	-	1
ST5824	singleton	1	-	1
ST5825	CC130	-	1	1
ST5844	CC425	-	1	1
ST5845	CC130	-	1	1
Total		140	104	244

**Table 5 animals-10-01109-t005:** Relation among the clonal complex (CC) isolates from rabbits with lesion/s and *S. aureus* carriers.

Lesions	Rabbits	CC Lesion	CC Nostril	CC Ear	CC Perineum
Dermatitis	OC19006	CC130/CC425	CC425	-	CC425
Conjunctivitis	OC19190	CC130	CC130	CC130	CC130
Hepatitis	OC19275	CC130	CC130	CC130	CC130
Hepatitis	OC19276	CC130	CC130	CC130	CC130

**Table 6 animals-10-01109-t006:** Characteristics of the MRSA isolates recovered in the different studied high-density towns.

Strain	Town	Anatomical Location	Molecular Typing				SCC*mec*	IEC	Antimicrobial Resistance Phenotype	Antimicrobial Resistance Genes
			*coa*/*spa*	MLST	CC	*agr*				
1439	Alfafara	Ear	B4 I1	ST1945	CC130	III	XI	*scn-sak* (group E)	PEN-FOX	*mec*C-SCC*mec*XI
1490	Cabanes	Dermatitis	B4 I1	ST5822	CC130	III	XI	-	PEN	*mec*C-SCC*mec*XI
1564	Cabanes	Nostril	B4 I1	ST1945	CC130	III	XI	-	Susceptible	*mec*C-SCC*mec*XI
1768	Castelló de Rugat	Ear	B4 I1	ST1945	CC130	III	XI	-	PEN-FOX	*mec*C-SCC*mec*XI
1657	Cheste	Ear	D1 I6	ST398	CC398	I	V	-	PEN-FOX-ENO-TET	*mec*A-*bla*Z-SCC*mec*V
1660	Cheste	Ear	B4 I1	ST5823	CC130	III	XI	-	PEN-FOX	*mec*C-SCC*mec*XI
1651	Cheste	Perineum	B4 I1	ST1945	CC130	III	XI	-	PEN-FOX	*mec*C-SCC*mec*XI
1867	Faura	Ear	B4 II4	ST1945	CC130	III	XI	-	FOX	*mec*C-SCC*mec*XI
1707	Godelleta	Nostril	B4 I1	ST1945	CC130	III	XI	-	PEN-FOX	*mec*C-SCC*mec*XI
1712	Godellerta	Ear	A1 IV1	ST5824	singleton	II	XI	-	PEN-FOX	*mec*C-SCC*mec*XI
1620	Montaverner	Ear	B4 I1	ST1945	CC130	III	XI	-	PEN-FOX	*mec*C-SCC*mec*XI
1468	Pedralba	Perineum	B4 II4	ST1945	CC130	III	XI	-	PEN-FOX	*mec*C-SCC*mec*XI
1802	Vilamarxant	Nostril	B4 I1	ST1954	CC130	III	XI	-	PEN-FOX	*mec*C-SCC*mec*XI
1272	Vinaròs	Dermatitis	B4 I1	ST1945	CC130	III	XI	*scn-sak* (group E)	PEN-FOX	*mec*C-SCC*mec*XI

PEN, penicillin; FOX, cefoxitin; ENO, enrofloxacin; TET, tetracycline.

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
