# Peer review of "Marked Presence of Methicillin-Resistant Staphylococcus aureus in Wild Lagomorphs in Valencia, Spain"

_animals, 2020, doi:10.3390/ani10071109_

Round 1
Reviewer 1 Report
In their manuscript “Marked presence of methicillin-resistant Staphylococcus aureus in wild lagomorphs from areas with rabbit high density”, authors showed the epidemiological implications by identified and characterized S. aureus, specifically MRSA, strains in wild lagomorph high-density areas. This study is meaningful and will help people to avoids the risks for the human health because these animals may enter the food chain. There are many aspects that should be improved:
- Hares and wild rabbits were collected from 14 high-density towns to check the presence of S. aureus, but there were only 10 districts showed in the Figure 1.
- Authors should show Ethics statement in the manuscript.
- Authors should show the PCR primers which used in the study.
- The abbreviation of “MSSA” should be defined.
- It’s better to show the gel image of PCR product for the detection of the IEC Cluster (scn, chp, sak and sea), blaZ, tst and PVL genes.
Author Response
There are many aspects that should be improved:
- Hares and wild rabbits were collected from 14 high-density towns to check the presence of S. aureus, but there were only 10 districts showed in the Figure 1.
The 14 towns are within the 10 districts studied, given the small size of Figure 1 (that shows the 10 districts) it could be confusing to also include the number corresponding to the 14 towns; Table 2 indicates the towns that belong to each district. If the reviewer and editor consider it essential, we could try to introduce the identification of towns and districts on the map, but we think it will be confusing.
- Authors should show Ethics statement in the manuscript.
We have specified in Material and methods (line 113) the origin of the animals with the following phrase: ‘Animals were donated by hunters for this study once they were dead`. As they are dead animals, they are not affected by animal welfare regulations or ethics statement.
- Authors should show the PCR primers which used in the study.
Table 1 has been added with PCR primers.
- The abbreviation of “MSSA” should be defined.
Done (introduction line 70).
- It’s better to show the gel image of PCR product for the detection of the IEC Cluster (scn, chp, sak and sea), blaZ, tst and PVL genes.
Figure 2 has been added with PCR products.
Reviewer 2 Report
I found this to be a well presented manuscript. The standard of English was high and results were clearly presented. If I have a criticism of the writing, it is that the manuscript is longer than it needs to be for the amount of information obtained in the project.
The title of the manuscript refers to rabbits in Valencia where density is high but although the authors presented a map showing relative densities of rabbits, no attempt seems to have been made to present data showing how the prevalence of methicillin-resistant S. aureus varied according to rabbit density. I understand that sampling was not adequate in several planned sites, but to me, this would seem a useful first step in determining whether rabbit density may be an important variable. The data could be presented as a simple scatter plot (S. aureus prevalence vs rabbit density index) showing sample size at each location and the fitted regression might be weighted accordingly. It would also be useful to have comparable information on rabbit density in Aragon where S. aureus has also been detected in rabbits or even other parts of Spain so that the reader can understand that rabbits are at very high abundance in the region sampled.
If there is no evidence suggesting a relationship between methicillin-resistant S. aureus prevalence and rabbit density, then reference to a high density population should be left out of the title of the manuscript, i.e. instead of saying 'with rabbit high density' simply say 'in Valencia, Spain.'
If the virus was spread in contaminated water, for example, rabbit density would probably be of little importance. Implying prevalence is density-related could be misleading if there is no clear evidence that is so.
Author Response
If I have a criticism of the writing, it is that the manuscript is longer than it needs to be for the amount of information obtained in the project.
The title of the manuscript refers to rabbits in Valencia where density is high but although the authors presented a map showing relative densities of rabbits, no attempt seems to have been made to present data showing how the prevalence of methicillin-resistant S. aureus varied according to rabbit density. I understand that sampling was not adequate in several planned sites, but to me, this would seem a useful first step in determining whether rabbit density may be an important variable. The data could be presented as a simple scatter plot (S. aureus prevalence vs rabbit density index) showing sample size at each location and the fitted regression might be weighted accordingly. It would also be useful to have comparable information on rabbit density in Aragon where S. aureus has also been detected in rabbits or even other parts of Spain so that the reader can understand that rabbits are at very high abundance in the region sampled.
If there is no evidence suggesting a relationship between methicillin-resistant S. aureus prevalence and rabbit density, then reference to a high density population should be left out of the title of the manuscript, i.e. instead of saying 'with rabbit high density' simply say 'in Valencia, Spain.'
If the virus was spread in contaminated water, for example, rabbit density would probably be of little importance. Implying prevalence is density-related could be misleading if there is no clear evidence that is so.
The manuscript (mainly discussion) has been shortened following reviewers’ indications. If you think there is still a need to shorten it further, please let us know and we will try to do so.
The authors agree that a study has not been conducted to demonstrate the relevance of density to relate it to the prevalence of S. aureus. Reference to a high-density population has been removed from the title of the manuscript and replaced by in Valencia, Spain. The density of rabbits is not the same in all areas of Valencia. As an initial hypothesis, it was thought that the interaction between rabbits and other animals is higher in high-density areas. Therefore, it was decided to sample regions with high rabbit density, and we believe that it is important to leave it in the manuscript. However, its importance in the discussion has been diminished.
Reviewer 3 Report
A large number of MRSA were identified among the tested strains from wild lagomorphs. This result is a relevant epidemiological finding also considering its possible implications on public health. Although wild species come infrequently in contact with humans, the study was carried out testing strains from hunted species which could represent a potential risk for humans also because of the minor hygienic measures often without veterinary control in slaughter in respect to the commercial rabbits.
Although the research design is appropriate and the manuscript well organized, I make some remarks:
Simple summary section
Line 25-26: the sentence sounds inexact. The presence of S. aureus is expected in samples from lagomorphs, also from healthy individuals, while MRSA are unexpected because wild rabbits have no direct contact with antibiotics. Please, rewrite the sentence.
Materials and Methods section
Which was the origin of the studied animals? Were they referred from hunters or were they found already dead in the hunting territory? Please, specify it in Animals and Geographical Locations paragraph (2.1.1.)
Results section
Line 183- 185: This information is more appropriate for Materials and Methods section. Therefore, it should be transferred.
Line 230-231: It is not clear the meaning of this sentence. I think that the Authors suggest that in a single case, corresponding to OC19006, a strain different from those from rostril/perineum was identified in the lesion. Please, rewrite the sentence more clearly.
Line 290-291: Likewise, the sentence is not clear and appears grammatically not correct. Please, rephrase it.
Discussion section
Some parts of Discussion section are redundant and repetitive. Sometimes, the obtained results are uselessly described and specified (lines 293-295, for example).
The introducing sentence (lines 250-252) is not really proper for the discussion while it is more suitable for Introduction section.
Author Response
Although the research design is appropriate and the manuscript well organized, I make some remarks:
Simple summary section
Line 25-26: the sentence sounds inexact. The presence of S. aureus is expected in samples from lagomorphs, also from healthy individuals, while MRSA are unexpected because wild rabbits have no direct contact with antibiotics. Please, rewrite the sentence.
We have modified the sentence in lines 25-26: “These are surprising results given the unexpected high presence of MRSA in the studied animals”.
Materials and Methods section
Which was the origin of the studied animals? Were they referred from hunters or were they found already dead in the hunting territory? Please, specify it in Animals and Geographical Locations paragraph (2.1.1.).
We have added the sentence “Animals were donated by hunters for this study once they were dead” in line 113 to clarify this point.
Results section
Line 183- 185: This information is more appropriate for Materials and Methods section. Therefore, it should be transferred.
The sentence in line 183-185 has been rewritten and transferred to Materials and Methods section, lines 110-113: “The 10 hares of the study were included because they were hunted together with rabbits and it was considered interesting to compare the findings observed in another lagomorph from the same geographical locations. Animals were donated by hunters for this study once they were dead”.
Line 230-231: It is not clear the meaning of this sentence. I think that the Authors suggest that in a single case, corresponding to OC19006, a strain different from those from nostril/perineum was identified in the lesion. Please, rewrite the sentence more clearly.
The sentence in lines 240-242 has been rewritten: “In a single case, corresponding to OC19006, a strain different from those from nostril/perineum was identified in the lesion”.
Line 290-291: Likewise, the sentence is not clear and appears grammatically not correct. Please, rephrase it.
The sentence in lines 303-304 has been rewritten: ”Only in one case, corresponding to rabbit OC19006, a strain different from those from nostril/perineum was identified in the lesion”.
Discussion section
Some parts of Discussion section are redundant and repetitive. Sometimes, the obtained results are uselessly described and specified (lines 293-295, for example).
We have shortened the sentence on the lines 306-307: ”The most frequently detected CC herein was CC130, followed by CC425”’ And the length of the discussion has been reduced.
The introducing sentence (lines 250-252) is not really proper for the discussion while it is more suitable for Introduction section.
The sentence in lines 250-252 has been removed.
Reviewer 4 Report
General comments
This manuscript is interesting and overall well presented and well written.
However, some improvements can be made.
Why methicillin or oxacillin (usually used as an equivalent to detect MRSA) was/were not added to your list of antibiotics in the antibiotic susceptibility testing?
Please revise Table 1. My understanding is that you presented the total of the animals tested (“Total”) then the number of animals positive for S. aureus at least at one on the sampling site (“S. aureus”). I do not understand the mecA and mecC column…If S. aureus was found at different localization what sample was considered in table 1?
The result section presenting resistance profile should be improved. The text is confusing. A table presenting the resistance profile of the MSSA and MRSA could help.
Please replace in your section titles S. Aureus for S. aureus.
Specific comments
- Abstract
L42 (“all isolates were fully susceptible to all the tested antibiotics”) seems contradictory to L45 (“ of the 28 MSSA strains showing resistance to penicillin”)… Please clarify.
- Keywords: I suggest to replace MecA for MecC
- Materials and Methods:
L127-129 : please clarify this sentence… Do you mean that the 2 towns are (1) in low and medium density districts, (2) located on the border of very high or high-density districts (3) also host high densities. It is not clear to what the last part of the sentences (3) refers to.
L153, 154: Please precise that your detection of the MRSA was done by PCR detection of the mecA or C genes and not the testing of their resistance to methicillin or any equivalent (oxacillin)
- Results
Antibiotic resistance profile section
Again in this section, the same confusion is present about “all the strains were fully susceptible to all the tested antibiotics, except MRSA” (L235) and ”28 strains resisted penicillin” (L238-239).
Please clarify this point.
L238: How were the % calculated ?? If it is 0/104 =0%.... Please precise for clarity
L239: What does it mean to be fully resistant ?
- Discussion
L267: I would suggest to replace “infect” by “colonize”
L268-269: Do you mean “if only one anatomical site is selected for sampling”
L278-279: Please be more cautious in this sentence because only 8 rabbits with lesions were examined
L356: Please italicize mec
L355-368: this paragraph is interesting but is it really relevant for this project where only 1 mecA isolate was observed ?
- Table 1: Please replace “total” by “number of animals tested”. Please replace S. aureus by number of animals positive for S. aureus” in the title of the column and precise as a footnote that they are positive at one or more of the sampling sites.
- Table 2 please replace “samples” by “animals tested”. Please clarify the title too. Suggestion “according to sampling localization)
Figure 1: Very nice and very informative but would it be possible to add the 2 towns of Pedralba and Vinaròs to the map to help the reader not familiar with Spain geography
Author Response
General comments
However, some improvements can be made.
Why methicillin or oxacillin (usually used as an equivalent to detect MRSA) was/were not added to your list of antibiotics in the antibiotic susceptibility testing?
We consider that the most reliable test to determine the strains that are MRSA is the detection of the mecA / mecC genes by PCR. To evaluate methicillin resistance with the antibiograms, we took the cefoxitin disc as a reference, because it is described that the antibiotic-resistant strains of cefoxitin are MRSA.
Please revise Table 1. My understanding is that you presented the total of the animals tested (“Total”) then the number of animals positive for S. aureus at least at one on the sampling site (“S. aureus”). I do not understand the mecA and mecC column…If S. aureus was found at different localization what sample was considered in table 1?
We have changed the titles in this table (now Table 2) to make them clearer. An animal that has a strain carrying this gene at either location is considered positive for mecA / mecC, even if an MSSA strain is isolated at another location.
The result section presenting resistance profile should be improved. The text is confusing. A table presenting the resistance profile of the MSSA and MRSA could help.
This section has been revised and the percentages have been removed because they were wrong. We think that now the results are more accurate without introducing a table. If the reviewer still considers that a table needs to be entered, please let us know.
Please replace in your section titles S. Aureus for S. aureus.
Done.
Specific comments
- Abstract
L42 (“all isolates were fully susceptible to all the tested antibiotics”) seems contradictory to L45 (“ of the 28 MSSA strains showing resistance to penicillin”)… Please clarify.
The sentence has been rewritten (lines 42-44): “Penicillin resistance was detected in 28 mec negative isolates and in one case bacitracin resistance. mecA isolate presented resistance to enrofloxacin and tetracycline, and 10 mecC isolates also showed bacitracin resistance”.
- Keywords: I suggest to replace MecA for MecC.
Done.
- Materials and Methods:
L127-129 : please clarify this sentence… Do you mean that the 2 towns are (1) in low and medium density districts, (2) located on the border of very high or high-density districts (3) also host high densities. It is not clear to what the last part of the sentences (3) refers to.
The sentence in lines 131-133 has been rewritten: “Samples from the towns of Pedralba and Vinarós were included in this study, which belong to low- and medium-density districts, respectively, and are located on the borders of very high- or high-density districts, respectively, that also host high densities due to the proximity to these districts”.
L153, 154: Please precise that your detection of the MRSA was done by PCR detection of the mecA or C genes and not the testing of their resistance to methicillin or any equivalent (oxacillin).
This sentence has been added in lines 158-159: “Isolates were classified into MRSA or MSSA according to the presence or absence results of the mecA/mecC genes obtained by PCR”.
- Results
Antibiotic resistance profile section
Again in this section, the same confusion is present about “all the strains were fully susceptible to all the tested antibiotics, except MRSA” (L235) and ”28 strains resisted penicillin” (L238-239). Please clarify this point.
The text has been revised and rewritten (lines 246-248): “Penicillin resistance was detected in 28 mec negative isolates and in one case bacitracin resistance. mecA isolate presented resistance to enrofloxacin and tetracycline, and 10 mecC isolates also showed bacitracin resistance”. We hope it is clearer now.
L238: How were the % calculated ?? If it is 0/104 =0%.... Please precise for clarity.
This section has been revised and the percentages have been removed because they were wrong.
L239: What does it mean to be fully resistant?
We have removed “fully” throughout the text.
- Discussion
L267: I would suggest to replace “infect” by “colonize”
Done (line 282).
L268-269: Do you mean “if only one anatomical site is selected for sampling”.
The sentence in lines 283-285 has been rewritten: “This is an important issue because it could affect the diagnosis of this condition if only one colony is selected during the microbiological identification of staphylococcal infections or only one anatomical site is selected for sampling”.
L278-279: Please be more cautious in this sentence because only 8 rabbits with lesions were examined.
The phrase has been removed as it is not relevant to the study due to the low number of samples.
L356: Please italicize mec.
This part of the discussion has been rewritten. We have checked that all mec genes are in italics.
L355-368: this paragraph is interesting but is it really relevant for this project where only 1 mecA isolate was observed ?
This paragraph has been removed as not relevant.
- Table 1: Please replace “total” by “number of animals tested”. Please replace S. aureus by number of animals positive for S. aureus” in the title of the column and precise as a footnote that they are positive at one or more of the sampling sites.
We have changed the titles and a footnote has been included in this table (now Table 2).
- Table 2 please replace “samples” by “animals tested”. Please clarify the title too. Suggestion “according to sampling localization).
In this table (now Table 3) we do mean samples and not positive animals. The title has been modified.
Figure 1: Very nice and very informative but would it be possible to add the 2 towns of Pedralba and Vinaròs to the map to help the reader not familiar with Spain geography.
The 14 towns are within the 10 districts studied, given the small size of Figure 1 (that shows the 10 districts) it could be confusing to also include the number corresponding to the 14 towns; Table 2 indicates the towns that belong to each district. If the reviewer and editor consider it essential, we could try to introduce the identification of towns and districts on the map, but we think it will be confusing.
Round 2
Reviewer 4 Report
Thank you for the revised manuscript and for the modifications that improved its quality.
No further changes are required.